# From words to worries: A cross-cultural comparison of parent-child conversations about snakes in early childhood

Reider Lori B.[1]*, Landry Yuan Félix[2,3], Leung Even Y.M.[3], Yeung Karen K.L.[4], Leung Samantha Hing Lam[4], Hai Catherine Wai Ching[3], Lei Janet Hiu Ching[5], LoBue Vanessa[1]

**1** Department of Psychology, Rutgers University, Newark, New Jersey, United States of America, **2** Institute for Resources, Environment and Sustainability, University of British Columbia, Vancouver, BC, California, United States of America, **3** School of Biological Sciences, The University of Hong Kong, Hong Kong SAR, People's Republic of China, **4** Faculty of Science, The University of Hong Kong, Hong Kong SAR, People's Republic of China, **5** Department of Psychiatry, The University of Hong Kong, Hong Kong SAR, People's Republic of China

☯ Reider Lori B. and Landry Yuan Félix contributed equally and share first-authorship.
* lori.reider@rutgers.edu

## Abstract

Fear of animals is shaped by psychological, cultural, and evolutionary influences over time. This study examined whether conversations about commonly feared animals—namely snakes—differ among parents and children from different cultures. Parents and children from the United States (n = 31; 28 Mothers, 2 Fathers, 1 Other; 15 Males, 16 Females, $M_{age}$ = 71.55 months, $SD_{age}$ = 12.56, $Range_{age}$ = 42) and Hong Kong (n = 31, 29 Mothers, 2 Fathers; 17 Males, 14 Females, $M_{age}$ = 76.48 months, $SD_{age}$ = 12.12, $Range_{age}$ = 44) read through a picture book of snakes, spiders, lizards, and turtles, and their conversations were coded for emotional content. Parents and children also completed several questionnaires, including a measure of their fear beliefs toward each animal. Parents and children across sites were more fearful of and used more negative language about snakes and spiders than lizards and turtles, and parents from Hong Kong expressed more fear and used more negative language about snakes than parents from the United States. However, despite more fear and negative language from Hong Kong parents, children from Hong Kong reported being *less* fearful of snakes (and spiders) than children from the United States, while Hong Kong children reported feeling *more* afraid of turtles and lizards. These findings highlight the prevalence of snake fears and negative information about snakes across different regions of the world, and potential differences in the kinds of input parents convey to children about snakes across cultures.

**Data availability statement:** The data underlying the results presented in the study are available from Open Science Framework using the following link: https://osf.io/2rp4j/overview?view_only=ecab827fdd2f4fb8a676dd58ff-260daa.

**Funding:** The current research was funded by a grant by the James S. McDonnell Foundation to Vanessa LoBue. The funders had no role in study design, data collection and analysis, decision to publish, or preparation of the manuscript.

**Competing interests:** The authors have declared that no competing interests exist.

## Introduction

Relationships between humans and other living creatures, particularly involving species posing potential threats to people, are diverse and can vary across social, cultural, and geographical contexts [1,2]. Snakes, for example, have historically represented evil and fear in many Western cultures, and are the targets of some of the most common fears and phobias throughout the world [3], likely because they represent threats to safety or survival [4]. In fact, snake fears are so common that researchers have suggested they might have an evolutionary origin, endowing humans with a predisposition to not only detect snakes very quickly, but to also rapidly acquire a long-lasting fear of these animals on first exposure (see [1] for a review). Indeed, several studies have demonstrated that children, adults, and non-human primates detect snakes faster than other kinds of animals (e.g., [5–8]) and adults form long-lasting associations between snakes and an aversive outcome (e.g., electric shock [1]).

For most modern humans, however, particularly in temperate, urban settings, fear of snakes is more widespread than the actual threat they pose. While snake envenomation is common globally, fatal encounters are rare, with an estimated 1.2 to 5.5 million people envenomed by snakes and about 125,000 (2% −10%) deaths annually [9]. Snake bite envenomation is a significant health concern for many marginalized societies in the tropics, especially in agricultural and rural areas [10], warranting its classification as a neglected tropical disease [11]. Asia and Africa, as well as lower income countries, are afflicted by snake bite morbidity and mortality to a higher degree than other continents or higher income countries [9]. However, while snake-bites continue to be a concern throughout the Global South [11], encounters with snakes that lead to injury or death are relatively rare compared to the global population (e.g., 5.5 million envenomation corresponds to 0.068% of the 8.03 billion global population). Even within these broad regions of the world, the prevalence of encounters with snakes varies based on ecological relevance and livelihood. Thus, while some snake species do constitute a threat to modern-day humans in many parts of the world, snake fears remain widespread across the globe regardless of experiential or ecological relevance to humans.

Given the narrow geographical scope of threats by snakes, why are they so widely feared? Fears can be acquired through several pathways, including direct experiences, vicarious observation, the transmission of negative information, or a combination of the three [12,13]. Given that snake bites are relatively uncommon globally, snake fears likely do not result from a direct negative experience for most individuals. However, there are several studies demonstrating that from a young age, children are exposed to a significant amount of negative information about snakes from books, media, and most importantly, from caregivers. In fact, the vast majority (89%) of intense childhood fears reported in the United States and Australia comes from threatening verbal information (e.g., negative information from parents) or seeing something threatening through media [14]. In one study where preschool-aged children walked through a reptile house at a local zoo with a parent, researchers reported that children were more likely to hear negative information about snakes from their parents when

compared to other, even perceptually similar animals, like lizards, frogs, and turtles [15]. In a similar study, preschool-aged children were given a simple storybook containing information about where different snakes, spiders, turtles, frogs, and lizards live, and parents were asked to simply read through the book with their children. Parents used significantly more negative information about snakes and spiders than turtles, frogs, and lizards, and significantly less positive information about snakes and spiders than the other animals as well [16]. This suggests that negative information about snakes (and spiders) is quite common in parent-child interactions in the US and may serve as a source of learning to fear these animals.

To date, most of the studies examining fear-learning of snakes has come from samples of largely White, middle-class families from Western high-income countries, where life-threatening encounters with snakes are extremely rare [15–17]. This is important, as the extent to which certain animals are feared can differ substantially across countries and cultures [18]. For example, as discussed above, the likelihood of negative encounters with snakes in particular is not universal and depends on socio-economic and environmental factors [10,19]. Accordingly, how snakes are discussed between parents and children, and whether or not fear develops, may depend on how threatening they are in the local environment [20], which itself varies across cultures and ecosystems.

Importantly, while snake fears are prevalent across the globe, fear of snakes may coexist with deeply rooted cultural values and practices. For example, diverse cultural practices and traditions centered on snakes exist throughout the world, and especially across Asia, where snake bite incidence and mortality rates are the highest globally [9]. Some are devotional, as in Southwest India, where snake deities are worshiped and snake killings discouraged, often for fear of being cursed and eventually bitten by a snake [21]. Others express utilitarian views, such as in Hong Kong and much of Southern China, where snakes are traditionally consumed as soup for medicinal and nutritional purposes [22,23], and their skin used in making traditional musical instruments [24,25]. In Chinese culture, metaphorical references can also reflect the snake's importance and object of awe in the Chinese zodiac and other myths [26,27]. These belief systems and traditions contrast with common, non-indigenous American experiences with snakes often limited to captive settings, such as zoos or as pets, as well as child directed books and media [15,28]. Comparatively, these latter, curated experiences with snakes are gaining popularity in Hong Kong (e.g., [29]), in addition to utilitarian traditions. Overall, both the US and Hong Kong share complex, and often negative, associations with snakes [26,30], but with socio-cultural circumstances, could evoke disparate thoughts and emotions in children. However, we still know very little about whether conversations about snakes, and the resulting influence on children's fear of snakes, differ across those cultures, where experiences and expectations of snakes differ.

The current study aimed to examine parent-child conversations about snakes and fear of snakes across cultures, with the inclusion of spiders as a commonly feared, yet relatively less medically or culturally salient animal for comparison. We aimed to examine differences among children and caregivers from the United States (US) and Hong Kong (HK). We chose these two regions because they reflect different worldviews about, and may have potentially different experiences with snakes, providing a meaningful basis to examine how cultural contexts may influence parent-child conversations about snakes, and animals more broadly. We asked whether parents and children report greater fear of snakes and another commonly feared animal—spiders—when compared to lizards and turtles, which are similar to snakes, but not commonly feared, and whether this differs by cultural region (US, HK), particularly for snakes. We also asked whether parents and children provide more negative and less positive information during conversations about snakes and spiders than lizards and turtles, and whether this differs by region. Finally, we explored relations between parents' and children's fear of each animal, their use of negative language about each animal, and whether parents' use of negative language about snakes and spiders predicted children's fear of those animals, including these same relations by site. We also explored whether children's frequency of nature visits were associated with children's fear beliefs and their use of negative language about each animal.

As discussed above, people in Asia are afflicted by snake bite morbidity and mortality to a higher degree than other continents or higher income countries like the US [9]. Further, there is evidence that adults from Hong Kong are more

fearful of typically fear-evoking animals when compared to the US [18]. Thus, we predict Hong Kong parents and children to use more negative language about snakes in particular, and express more fear towards snakes than US adults and children. The same might not be true for lizards, turtles, or even spiders. Understanding cultural differences in parent-child discussions about commonly feared animals can clarify how negative language shapes children's developing animal fears and, more broadly, the cultural and environmental influences on fear development.

## Materials and methods

The original data collection and analysis plan included parent-child dyads from 3 locations: United States (US), Hong Kong (HK), and India, but due to logistical constraints, we were only able to move forward with two sites (US, HK) and therefore all analyses were adjusted accordingly (see analysis plan). All data were collected during an online Zoom call using Qualtrics during the COVID-19 pandemic, from February 9 through May 26, 2022.

## Participants

Participants were recruited for an online study of parent-child interactions during a picture book reading task. Participants were eligible if they 1) lived in the United States or Hong Kong and 2) self-reported their child was between the ages of 4 and 6 years old at the time of sign up for the study. This age range was selected as this is around the time when animal fears and phobias first emerge in childhood [31]. The final sample included 62 parents (57 Mothers, 4 Fathers, 1 Other) and children (32 Males, 30 Females, $M_{age}$ = 74.02 months, $SD_{age}$ = 12.49, $Range_{age}$ = 55), including 31 parent-child dyads from the United States and 31 dyads from Hong Kong. Sample size was based on previous research using similar methodologies with parents and children (e.g., [15,16]). Sites did not significantly differ by child age, $t(60)$ = 1.57, $p$ = .12 or by child sex, $χ^2(1)$ =.06, $p$ = .80. Below we describe demographic information for each sample separately, as there were slight differences in how questions were asked to align with data collection at each site (e.g., changing the currency for household income).

**Hong Kong (HK) sample.** The HK sample included 31 parents (29 Mothers, 2 Fathers) and children (17 Males, 14 Females, $M_{age}$ = 76.48 months, $SD_{age}$ = 12.12, $Range_{age}$ = 44). Parents described their neighborhood as rural (n = 2, 6.45%), suburban (n = 11, 35.48%), or urban (n = 18, 58.06%). All parents self-identified as Ethnic Chinese (n = 31, 100.00%) and identified their children as Ethnic Chinese (n = 31, 100.00%). Parents reported having a bachelor's degree (n = 14, 45.16%), masters or above (n = 10, 32.26%), bachelors and masters or above (n = 1, 3.23%), diploma (non-bachelor) n = 1, 3.23%), diploma (non-bachelor) and bachelors (n = 1, 3.23%), or high school, F4-F7 (n = 4, 12.90%). Parents also reported their average household income in the last three years as USD $100,000 (HKD $780,000 or more) (n = 14, 45.16%), USD $60,000- $100,000 (HKD $470,000-$780,000) (n = 8, 25.81%), USD $20,000- $40,000 (HKD $160,000- $310,000) (n = 5, 16.13%), USD $40,000- $60,000 (HKD $310,000- $470,000) (n = 3, 9.68%), or USD $20,000 (HKD $160,000) (n = 1, 3.23%). Children heard Cantonese only (n = 24, 77.42%), Cantonese and another language (n = 3, 9.68%) or Cantonese and two other languages (n = 4, 12.90%) in the home. Parents reported their children as having never lived with a pet (n = 18, 58.06%), currently living with a pet (n = 10, 32.26%), or have previously lived with a pet (n = 3, 9.68%). None of the children currently or have ever lived with any of the animals included in the study.

**United States (US) sample.** The US sample included 31 parents (28 Mothers, 2 Fathers, 1 Other) and children (15 Males, 16 Females, $M_{age}$ = 71.55 months, $SD_{age}$ = 12.56, $Range_{age}$ = 42). Parents described their neighborhood as rural (n = 3, 9.68%), suburban (n = 22, 70.97%), or urban (n = 6, 19.35%). Parents self-identified as White, not of Hispanic origin (n = 22, 70.97%), Asian/Pacific Islander (n = 5, 16.13%), more than one race/ethnicity (n = 2, 6.45%), American Indian/Alaska Native (n = 1, 3.23%), and one participant did not provide this information (3.23%). Parents identified their children as White, not of Hispanic origin (n = 21, 67.74%), more than one race/ethnicity (n = 5, 16.13%), Asian/Pacific Islander (n = 3, 9.68%), or American Indian/Alaska Native (n = 2, 6.45%). Parents reported having an advanced degree (n = 16, 51.61%), an AA/BA degree (n = 12, 38.71%), a high school diploma/GED (n = 1, 3.23%), some college/trade school

(n = 1, 3.23%), or some school no diploma (n = 1, 3.23%). Parents also reported their average household income in the last three years as more than $100,000 per year (n = 16, 51.61%), $40,000-$59,999 (n = 8, 25.81%), $60,000-$100,000 (n = 4, 12.90%), or $20,000-$39,999 (n = 3, 9.68%). Children heard English only (n = 24, 77.42%) or English and another language (n = 7, 22.58%) in the home. Parents reported their children as currently living with a pet (n = 19, 61.29%), having never lived with a pet (n = 6, 19.35%), or having previously lived with a pet (n = 6, 19.35%). Two children were reported to be currently living with a pet turtle and one child had previously lived with a pet turtle (no other children had ever lived with any of the animals included in the study).

## Materials

**Picture book.** All study materials were provided to participants using Qualtrics (Qualtrics, Provo, UT) during a live online Zoom session with a researcher. The researcher first provided instructions to parents about how to navigate the book. The picture book contained 16 pages of animals presented in a randomized order, each with a single neutral image of an animal, followed by a brief sentence with the animal category and where the animal lives (e.g., "This is a snake. This kind of snake can be found in Africa.") (see S1 Fig in S1 File for a sample page). The book included 4 categories of animals—snakes, spiders, lizards, and turtles—with one animal image per page and 4 different species represented from each animal category. Snakes were selected due to being highly feared animals with different regional cultural associations, while spiders were included due to their high fear rates but less prevalent cultural significance. Lizards and turtles were also included in the book as these are animals that do not commonly elicit fear. These animal categories have all been used in prior research as these animals are commonly found and discussed in similar settings that children are exposed to such as books about animals, trips to the zoo, or other settings (e.g., [15,16]).

Table 1 includes the 16 specific animals included in the picture book, all of which were used in a previous study [16]. Within each animal category we included 2 species of lower threat to humans and 2 species of higher threat (e.g., venomous) to humans to ensure the affective quality of the stimuli were as balanced as possible. All animal images were found

**Table 1. List of animal species included in the picture book and their threat relevance to humans.**

| Animal Category | Species | Threat Relevance to Humans |
|---|---|---|
| Snake 1 | Black Mamba | High |
| Snake 2 | King Cobra | High |
| Snake 3 | Garter Snake | Low |
| Snake 4 | Bull Snake | Low |
| Spider 1 | Brown Recluse | High |
| Spider 2 | Funnel Web Spider | High |
| Spider 3 | Jumping Spider | Low |
| Spider 4 | Huntsman Spider | Low |
| Lizard 1 | Gila Monster | High |
| Lizard 2 | Crocodile Monitor | High |
| Lizard 3 | Skink | Low |
| Lizard 4 | Bearded Dragon | Low |
| Turtle 1 | Alligator Snapping Turtle | High |
| Turtle 2 | Matamata turtle | High |
| Turtle 3 | Central American Wood Turtle | Low |
| Turtle 4 | Galapagos Tortoise | Low |

through Google searches or from Zoo websites and included a full body image of each real animal, pictured in isolation and in a neutral position (i.e., not eating or attacking) in their natural habitat.

**Children's fear beliefs.** Children completed a modified version of the Fear Beliefs Questionnaire (FBQ; [32]), which consisted of 28 questions including seven items (four reverse scored) regarding children's beliefs about each animal category using a 5-point Likert scale presented visually, ranging from 1- *no not at all* to 5- *yes, definitely.* Items were read aloud to the child by the researcher and recorded in Qualtrics during the study session. As was done in previous research using this measure (e.g., [16]), when children's responses did not match the exact item response (e.g., saying "yes" instead of "yes, probably" or "yes, definitely"), the researcher asked the child to specify between the two options on that side of the scale (e.g., yes probably or yes definitely?), and when a child was unable to make this distinction, the less extreme response was selected (yes, probably or no, not really). A fear beliefs score for each animal category was calculated as the average of the seven items asked about each animal, with higher scores indicating higher fear beliefs of the animal category.

**Parent knowledge and fear belief survey.** Parents completed a survey about their own fear and knowledge of the 16 animals presented in the picture book. For each animal, parents were asked "*Do you know what kind of [animal] this is*?" with the option to respond with "*Yes*" or "*No*" to assess their knowledge of the animal. If the parent's response was "*Yes*", they were asked to label the specific animal. As a measure of fear, parents were asked three items regarding the level of threat, fear, and willingness to approach each animal (reverse scored), and the average of the three items was used as a measure of parents fear beliefs (note the fear and willingness to approach items were the same as those asked in the child FBQ). For each item, parents indicated their fear beliefs using a 5-point Likert scale, ranging from 1- *no not at all* to 5- *yes, definitely*, in line with children's fear beliefs survey. A fear beliefs score was calculated by taking the average of all items asked about each animal category (3 items asked for 4 animal species per category, for a total of 12 items per animal category), with higher scores indicating higher fear beliefs about each animal category.

**Child experiences and demographics.** Parents were asked whether their child had ever seen or held/touched a live snake/spider/lizard/turtle, how often their child took part in nature visits or trips to the zoo or aquarium, and to report on any pet ownership during the child's life. Parents answered additional questions about their child's date of birth, child's biological sex, child's and parent's racial/ethnic background, parent's relationship to the child, neighborhood type (rural, suburban, urban), languages spoken at home, parent education, household income, and household size.

## Procedure

All procedures and materials were approved by the Institutional Review Board at Rutgers University (study title: "Learning, Perception, and Belief Revision in Infants, Children, and Adults," #Pro-2020000399) and the Human Research Ethics Committee at the University of Hong Kong (study title: "Learning, Perception, and Belief Revision Towards Snakes in Infants, Children, and Adults," #EA210495). Participants from the US were recruited through social media, our lab website, the *ChildrenHelpingScience.com* website, and word of mouth. Participants from the HK sample were recruited through word of mouth and through a family-targeted eco-tourism company (Little Woods Nature Education Limited).

The study took place using an online video call using Zoom. Parents and children completed the study from home with a researcher on the other end of the call. Parents provided verbal consent as well as completed an online consent form on behalf of themselves and their child. Parents and children were then asked to read the picture book together as they typically would read a book together. Following the instructions, the researcher turned off their camera and audio, so the picture book was the main activity on the participant's screen. Parents and children then read through the 16 pages of the book in a randomly presented order.

Following the picture book, the child completed the FBQ with the researcher. To begin, the researcher provided the child with instructions, including practicing how to answer questions using the response scale, and then verbally asked each question to each child. Children could respond to questions verbally (e.g., "yes, definitely") or physically (e.g., two

thumbs up) using a hand gesture scale with thumbs up and down options, consistent with previous research [17]. Items within each animal category block were presented in a randomized order, and the order of animal categories was also randomized. Following the FBQ, parents then completed the survey about their knowledge and fear of the animals, their child's experiences, and demographics. After completing the study, participants were debriefed about the nature of the study, and any questions were answered prior to the end of the session. In the US sample, families were compensated with a $10.00 Amazon gift card as a thank you for their time and participation. In the HK sample, families were compensated with a certificate of appreciation for their contributions to the study.

## Picture book conversation coding

Conversations during the picture book reading were transcribed as individual utterances provided by parents and children for the specific animal page on the screen during the discussion and were checked and finalized by a second researcher prior to coding. Conversations were then coded for positive, negative, or neutral information about the animals on each page using a coding scheme used in previous studies [16,17]. Utterances that were not directly related to the picture book discussion were not coded (e.g., statements about asking for snacks, statements about other animals or content not included in the book). Utterances could be coded as both positive and negative if the utterance referenced both positive and negative information (e.g., "kind of cute, kind of scary, and a little interesting"). Neutral statements were coded anytime utterances included only neutral information, including information that was read directly from the book, or when any questions or statements regarding the book content were provided, but were not emotional (e.g., discussions about what continent the animal lives on). Positive and negative utterances were further coded by the type of emotional language used but were not examined in this analysis.

At each site, a primary coder was trained and coded all the video transcripts. To establish interrater reliability, an additional coder was trained and independently coded 10 of the transcripts (US: n = 31, 32% double coded; HK: n = 31, 32% double coded). Cohen's Kappa (κ) was used to calculate reliability, with values between .60−.79 as moderate agreement, and values above .80 considered as strong agreement between raters [33]. We obtained an average Cohen's κ of .86 (US) and .80 (HK) for positive, negative, and neutral codes, indicating a very good level of agreement. The primary coder's data from each site was used for all analyses. After establishing reliability, the total number and proportion of positive, negative, and neutral utterances for each animal category (snakes, spiders, lizards, turtles) were calculated for each speaker (parent, child).

## Data analytic plan

The original analytic plan was preregistered on AsPredicted.org (#87741, https://aspredicted.org/7vfh-dzj8.pdf). However, because we were unable to collect data from one of the three sites due to logistical constraints, all planned analyses have been adjusted to account for this change (all analyses comparing sites compare two sites instead of three sites). Analyses were conducted using R version 4.3.2 [34].

We first describe children's experience with animals and parents' prior knowledge about the animals portrayed in the book. For our main and pre-registered analyses, we asked whether parents provided different amounts of positive and negative information about snakes and spiders than lizards and turtles across regions. Two separate 2 (site: US, HK) by 4 (animal: snake, spider, lizard, turtle) ANOVAs were conducted to assess differences in positive and negative information provided by parents across sites. We then asked whether children provided different amounts of positive and negative information about snakes and spiders than lizards and turtles across sites. Two separate 2 (site: US, HK) by 4 (animal: snake, spider, lizard, turtle) ANOVAs were conducted to assess differences in positive and negative information provided by children across sites. We also explored differences in use of neutral language for parents and children using separate 2 (site: US, HK) by 4 (animal: snake, spider, lizard, turtle) ANOVAs. We also explored these same ANOVAs using proportion scores (e.g., proportion of negative, positive, and neutral language used). Next, we asked whether there were

differences in parent and child fear beliefs based on the animal category and site. A separate 2 (site: US, HK) by 4 (animal: snake, spider, lizard, turtle) ANOVA was conducted for parents' and children's fear beliefs to examine whether there are differences in fear beliefs toward each animal across the sites. We explored whether parents' fear correlated with children's fear of the animals, whether parents' use of negative language correlated with children's use of negative language, and whether parents' use of negative language about snakes and spiders predicted children's fear of snakes and spiders. We also explored these same relations by site. Finally, we explored whether parent reports of children's frequency with nature visits was correlated with children's fear beliefs about each animal and their use of negative language about each animal.

## Results

### Children's experience with animals

In the parent survey administered through Qualtrics, parents were asked to report on their child's experience with animals and nature. First, parents were asked if their child had ever seen or held/touched a live snake, spider, turtle, or lizard snake (S1 Table in S1 File). The majority of children had seen each of the animals (n's = 46–59, 74.19% − 95.16%), with fewer children having ever held or touched each of the animals (n's = 19–34, 30.65% − 54.84%). Rates of having seen a snake (n = 46) or held/touched a spider (n = 12) were the lowest of all the animals, while rates of having seen (n = 59) or held/touched a turtle (n = 34) were highest of all the animals. There were some site variations in children's experiences with animals. For example, a greater number of children from the US have seen a snake (n = 26), and held or touched a snake (n = 11) than HK children (seen n = 20, held/touched = 8), with similar patterns for US and HK children having seen a spider (US n = 29, HK n = 24) and having held or touched a spider (US n = 7, HK n = 5).

Parents were also asked about how often their child takes part in activities where these animals may be seen (e.g., nature visits, visits to the zoo or aquarium). All parents reported that their children have participated in these activities at least occasionally, with more US parents selecting "very often" as the frequency than HK parents (S2 Table in S1 File). There were some site differences with respect to the frequency with which children engage in these activities. For example, more children from the US engaged in nature visits very often (n = 18) and zoo/aquarium visits very often (n = 6) than HK children (nature visits n = 3, zoo/aquarium n = 1).

### Parents' knowledge of animals

In the Qualtrics survey following the picture book task and child fear beliefs interview, parents were asked to identify each of the animal species presented in the book to assess prior knowledge (Table 2). In general, parents were able to accurately identify certain animal species, including the king cobra snake (n = 43, 69.35%), tortoise (n = 21, 33.87%), snapping turtle (n = 10, 16.13%), bearded dragon (n = 8, 12.90%) and garter snake (n = 7, 11.29%), with six other species accurately identified < 10% of the time and five that were not identified at all. There was some variation in parent's knowledge of animals by site. For example, more US parents (77.42%) accurately labeled the king cobra (the most accurately labeled animal from the book) than HK parents (61.29%). Some US parents also accurately identified the garter snake (22.58%), but no parents accurately identified the black mamba or bull snake. Accuracy rates were low for all spiders presented in the book and at similar rates across sites.

### Conversation analyses

On average, parents produced 99.02 utterances (SD = 68.11, Min = 12, Max = 422) and children produced 60.16 utterances (SD = 39.33, Min = 3, Max = 169) while reading the picture book together. There were no significant differences in the total number of utterances by site for parents, $t(60) = 1.14$, $p = .26$, or children $t(60) = .54$, $p = .59$. For the remaining conversation analyses on the total use of positive, negative, and neutral language, we removed 29 data points that were outliers as

Table 2. Parents' Prior Knowledge of Each Animal for the Overall Sample (n = 62) and by Site (HK, US, n = 31 for each).

| Animal Category | Species | Overall Sample | HK | US |
|---|---|---|---|---|
| | | Accurate n (%) | Accurate n (%) | Accurate n (%) |
| Snake 1 | Black Mamba | 0 (0%) | 0 (0%) | 0 (0%) |
| Snake 2 | King Cobra | 43 (69.35%) | 19 (61.29%) | 24 (77.42%) |
| Snake 3 | Garter Snake | 7 (11.29%) | 0 (0%) | 7 (22.58%) |
| Snake 4 | Bull Snake | 0 (0%) | 0 (0%) | 0 (0%) |
| Spider 1 | Brown Recluse | 1 (1.61%) | 0 (0%) | 1 (3.23%) |
| Spider 2 | Funnel Web Spider | 0 (0%) | 0 (0%) | 0 (0%) |
| Spider 3 | Jumping Spider | 2 (3.23%) | 1 (3.23%) | 1 (3.23%) |
| Spider 4 | Huntsman Spider | 5 (8.06%) | 0 (0%) | 5 (16.13%) |
| Lizard 1 | Gila Monster | 4 (6.45%) | 0 (0%) | 4 (12.90%) |
| Lizard 2 | Crocodile Monitor | 0 (0%) | 0 (0%) | 0 (0%) |
| Lizard 3 | Skink | 2 (3.23%) | 0 (0%) | 2 (6.45%) |
| Lizard 4 | Bearded Dragon | 9 (14.52%) | 0 (0%) | 9 (29.03%) |
| Turtle 1 | Alligator Snapping Turtle | 10 (16.13%) | 2 (6.45%) | 8 (25.81%) |
| Turtle 2 | Matamata turtle | 0 (0%) | 0 (0%) | 0 (0%) |
| Turtle 3 | Central American Wood Turtle | 1 (1.61%) | 0 (0%) | 1 (3.23%) |
| Turtle 4 | Galapagos Tortoise | 21 (33.87%) | 11 (35.48%) | 10 (32.26%) |

outlined in our preregistered plan. Outliers were defined across 48 variables as values that exceeded 3 standard deviations from the mean of language type (positive, negative, neutral) for each speaker (parent and child), site (US and HK), and animal (snakes, spiders, lizards, turtles). This included 6 US parent data points, 7 US child data points, 9 HK parent data points, and 8 HK child data points (only outlier values were dropped—participants remained in the dataset for all other analyses). We also calculated proportion scores, defined as the proportion of positive, negative, and neutral information about each animal and reran analyses examining differences in language by speaker and animal proportion scores. Using the same outlier procedure, 39 data points were removed including 8 US parent data points, 8 US child data points, 10 HK parent data points, and 13 HK child data points.

We primarily describe results of the total number of utterances because we believe that use of *any* emotional information may be influential on children's attitudes toward animals, but we also ran analyses with proportion scores to control for the amount that parents and children talked about each animal overall (e.g., a higher proportion of negative information about snakes from parents would indicate the information from parents was more negative about this animal, whereas the raw data will indicate the number of times something negative was referenced independent of the amount they said). Given the similarity in results, all proportion analyses are provided in the supplemental file and briefly described in the manuscript.

First, we ran separate 2 (site: US, HK) by 4 (animal: snakes, spiders, lizards, turtles) ANOVAs to examine whether parents and children provided different amounts of positive, negative, and neutral information. We ran these analyses for both the total scores and proportion scores. Descriptives are presented in S3 and S4 Tables in S1 File (total scores) and S5 and S6 Tables in S1 File (proportion scores) for parents' and children's language, respectively. All ANOVA analyses were conducted using the afex package version 1.4.1 [35] and the emmeans package version 1.10.6 [36] for post-hoc comparisons. All interaction effects were broken down using pairwise comparisons with Bonferroni adjusted p-values.

**Parent language analyses.** For parents' total use of negative language (Table 3 and Fig 1), we found a significant main effect of animal, $F(3, 159) = 18.25$, $p < .001$, $\eta^2 = .26$, with parents using significantly more negative language about snakes and spiders than lizards and turtles ($p$'s $< .001$, S7 Table in S1 File). We also found a significant site by animal

**Table 3. Total Utterances: Results for 2(site: US, HK) x 4(animal: snakes, spiders, lizards, turtles) ANOVA on Parents' Negative Language Use.**

| Predictor | df | F | P | η² |
|---|---|---|---|---|
| Between-Subjects | | | | |
| Site | 1 | 1.18 | .28 | .02 |
| Error | 53 | | | |
| Within-Subjects | | | | |
| Animal | 3 | 18.25 | <.001** | .26 |
| Site x Animal | 3 | 3.52 | .02* | .06 |
| Error | 159 | | | |

*p<.05 \* p<.001 \*\**.

Results are shown from a 2 (site: US, HK) × 4 (animal: snakes, spiders, lizards, turtles) ANOVA predicting the total number of parents' negative utterances. F values, degrees of freedom, p values, and η² are reported.

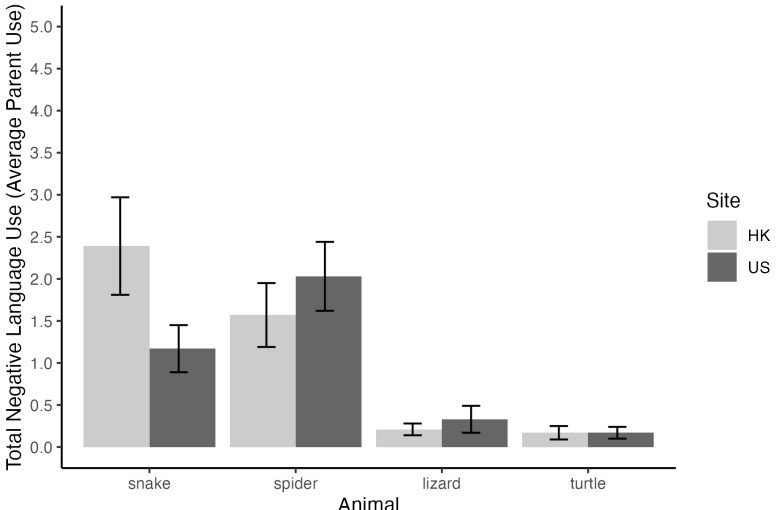

**Fig 1. Results for 2(site: US, HK) x 4(animal: snakes, spiders, lizards, turtles) ANOVA on Parents' Negative Language Use (total score data).**
Fig 1 represents parents use of negative language about each animal category included in the study. The Y axis represents the total number of negative utterances from parents, and the X axis represents each animal category. The bars represent the average score by site (HK or US) with standard error bars.

interaction, $F(3, 159) = 3.52$, $p = .02$, $η² = .06$ (S7 Table in S1 File). For between-site comparisons for each animal, HK parents used significantly more negative language about snakes than US parents ($p = .04$), consistent with our predictions. For within-site comparisons, HK parents used significantly more negative language about snakes than turtles ($p < .001$) and lizards ($p < .001$), and about spiders than turtles ($p = .003$) and lizards ($p = .01$). US parents used significantly more negative language about spiders than turtles ($p < .001$) and lizards ($p < .001$). For the proportion of negative language used by parents (S8 Table in S1 File), there was only a significant main effect of animal ($F(3, 162) = 21.40$, $p < .001$, $η² = .28$), with parents using a significantly greater proportion of negative language about snakes and spiders than lizards and turtles ($p$'s $< .001$, S9 Table in S1 File).

For parents' total use of positive language (Table 4), we found a significant main effect of animal, $F(3, 165) = 3.01$, $p = .03$, $η² = .05$, with parents using significantly less positive language about snakes than turtles ($p = .02$, S10 Table

**Table 4. Total Utterances: Results for 2(site: US, HK) x 4(animal: snakes, spiders, lizards, turtles) ANOVA on Parents' Positive Language Use.**

| Predictor | df | F | p | η² |
|---|---|---|---|---|
| Between-Subjects | | | | |
| Site | 1 | .83 | .37 | .01 |
| Error | 55 | | | |
| Within-Subjects | | | | |
| Animal | 3 | 3.01 | .03* | .05 |
| Site x Animal | 3 | 1.57 | .20 | .03 |
| Error | 165 | | | |

$p < .05$ * $p < .001$ **.

Results are shown from a 2 (site: US, HK) × 4 (animal: snakes, spiders, lizards, turtles) ANOVA predicting the total number of parents' positive utterances. F values, degrees of freedom, p values, and η² are reported.

in S1 File). When we ran the same analysis using the proportion of positive language, nothing was significant at $p < .05$ (S11 Table in S1 File). For parent's total use of neutral language (Table 5), we found a significant main effect of animal, $F(3, 177) = 2.93$, $p = .04$, $η² = .05$, with parents using less neutral language about snakes than lizards ($p = .01$, S12 Table in S1 File). When we ran the same analysis using the proportion of neutral language (S13 Table in S1 File), we found a significant main effect of animal $F(3, 174) = 7.61$, $p < .001$, $η² = .12$, with parents using a greater proportion of neutral language about lizards and turtles than snakes ($p$'s $< .02$) and about turtles than spiders ($p$'s $< .03$, S14 Table in S1 File).

**Child language analyses.** For children's total use of negative language (Table 6), we found a significant main effect of animal, $F(3, 168) = 15.01$, $p < .001$, $η² = .22$, with children using significantly more negative language about snakes and spiders than turtles ($p$'s $< .03$), about spiders than lizards ($p < .001$), and about spiders than snakes ($p = .02$, S15 Table in S1 File). When we ran the same analysis using the proportion of negative language, (S16 Table in S1 File), we found a significant main effect of animal, $F(3, 168) = 18.19$, $p < .001$, $η² = .25$, with children using a significantly greater proportion of negative language about snakes and spiders than lizards and turtles ($p < .05$, S17 Table in S1 File). We also found a significant site by animal interaction, $F(3, 168) = 3.92$, $p = .01$, $η² = .07$ (S17 Table in S1 File). For between-site comparisons for each animal, US children used a significantly greater proportion of negative language about spiders than HK children ($p = .01$), contrary to our predictions. For within-site comparisons for animals, US children used a significantly greater proportion of negative language about spiders than lizards, turtles, and snakes ($p < .01$) and about spiders than turtles ($p < .01$). No other comparisons were significant at $p < .05$, and there were no significant follow-up findings for the HK sample (S17 Table in S1 File).

**Table 5. Total Utterances: Results for 2(site: US, HK) x 4(animal: snakes, spiders, lizards, turtles) ANOVA on Parents' Neutral Language Use.**

| Predictor | df | F | p | η² |
|---|---|---|---|---|
| Between-Subjects | | | | |
| Site | 1 | .00 | .95 | .00 |
| Error | 59 | | | |
| Within-Subjects | | | | |
| Animal | 3 | 2.93 | .04* | .05 |
| Site x Animal | 3 | 2.21 | .09 | .04 |
| Error | 177 | | | |

$p < .05$ * $p < .001$ **.

Results are shown from a 2 (site: US, HK) × 4 (animal: snakes, spiders, lizards, turtles) ANOVA predicting the total number of parents' neutral utterances. F values, degrees of freedom, p values, and η² are reported.

**Table 6. Total Utterances: Results for 2(site: US, HK) x 4(animal: snakes, spiders, lizards, turtles) ANOVA on Children's Negative Language Use.**

| Predictor | df | F | P | η² |
|---|---|---|---|---|
| Between-Subjects | | | | |
| Site | 1 | 2.60 | .11 | .04 |
| Error | 56 | | | |
| Within-Subjects | | | | |
| Animal | 3 | 15.01 | <.001** | .22 |
| Site x Animal | 3 | 1.18 | .32 | .02 |
| Error | 168 | | | |

*p*<.05 * *p*<.001 **.

Results are shown from a 2 (site: US, HK) × 4 (animal: snakes, spiders, lizards, turtles) ANOVA predicting the total number of children's negative utterances. F values, degrees of freedom, p values, and η² are reported.

There were no significant effects for children's total use of positive language (Table 7), proportion of positive language (S18 Table in S1 File) at *p*<.05, or total use of neutral language (Table 8) at *p*<.05. For children's proportion of neutral language (S19 Table in S1 File), there was a main effect of animal, $F(3, 168) = 7.11$, $p<.001$, $η^2=.11$, with children using a significantly greater proportion of neutral language about turtles than spiders ($p<.01$, S20 Table in S1 File). We also found a significant site by animal interaction, $F(3, 168) = 2.99$, $p=.03$, $η^2=.03$. For between-site comparisons for each animal, HK children used a significantly greater proportion of neutral language about lizards ($p=.04$) and spiders ($p=.03$) than US children. For within-site comparisons for animals (S20 Table in S1 File), US children used a significantly greater proportion of neutral language about turtles than lizards and spiders ($p's<.03$). No other comparisons were significant at $p<.05$, and there were no significant follow-up findings for the HK sample (S20 Table in S1 File).

## Fear beliefs

Next, a 2 (site: US, Hong Kong) by 4 (animal: snakes, spiders, lizards, turtles) ANOVA was conducted for parents' and children's fear beliefs to examine differences in fear beliefs toward each animal and whether this differed by site. For the fear beliefs analyses, we removed 4 data points as outliers, which were data points defined as exceeding 3 standard deviations from the mean of fear beliefs for each speaker (parent and child), site (US and HK) and animal (snakes, spiders,

**Table 7. Total Utterances: Results for 2(site: US, HK) x 4(animal: snakes, spiders, lizards, turtles) ANOVA on Children's Positive Language Use.**

| Predictor | df | F | p | η² |
|---|---|---|---|---|
| Between-Subjects | | | | |
| Site | 1 | 1.17 | .28 | .02 |
| Error | 54 | | | |
| Within-Subjects | | | | |
| Animal | 3 | 1.34 | .26 | .02 |
| Site x Animal | 3 | 1.08 | .36 | .02 |
| Error | 162 | | | |

*p*<.05 * *p*<.001 **.

Results are shown from a 2 (site: US, HK) × 4 (animal: snakes, spiders, lizards, turtles) ANOVA predicting the total number of children's positive utterances. F values, degrees of freedom, p values, and η² are reported.

**Table 8. Total Utterances: Results for 2(site: US, HK) x 4(animal: snakes, spiders, lizards, turtles) ANOVA on Children's Neutral Language Use.**

| Predictor | df | F | p | η² |
|---|---|---|---|---|
| Between-Subjects | | | | |
| Site | 1 | .00 | .96 | <.001 |
| Error | 59 | | | |
| Within-Subjects | | | | |
| Animal | 3 | 2.25 | .08 | .09 |
| Site x Animal | 3 | 1.25 | .29 | .02 |
| Error | 177 | | | |

*p<.05 \* p<.001 \*\*.*

Results are shown from a 2 (site: US, HK) × 4 (animal: snakes, spiders, lizards, turtles) ANOVA predicting the total number of children's neutral utterances. F values, degrees of freedom, p values, and η² are reported.

lizards, turtles). This included 2 HK child data points, and 2 HK parent data points. Descriptives for parents' and children's fear of each animal category for the overall sample and by site can be found in Table 9.

For parents' fear of animals (Table 10 and Fig 2), we found a significant main effect of site, $F(1, 58) = 7.76$, $p = .01$, $η² = .12$, with HK parents reporting more fear than US parents overall, $t(58) = 2.79$, $p = .01$ (S21 Table in S1 File). We also found a significant main effect of animal, $F(3, 174) = 164.41$, $p < .001$, $η² = .74$, with parents reporting significantly greater fear of snakes than spiders, turtles, and lizards (all $p$'s < .001), significantly greater fear of spiders than turtles and lizards (all $p$'s < .001), and significantly greater fear of lizards than turtles ($p < .001$, S21 Table in S1 File). We also found a significant site by animal interaction, $F(3, 174) = 3.89$, $p = .01$, $η² = .06$ (S21 Table in S1 File). For between-site comparisons for each animal, HK parents reported significantly more fear of snakes ($p = .01$) and lizards ($p = .001$) than US parents. For within-site comparisons for animals, results showed similar patterns across sites. HK and US parents reported significantly more fear of snakes than spiders, lizards, and turtles (HK: $p$'s < .001; US: $p$'s < .01), significantly more fear of spiders than lizards and turtles (HK: $p$'s < .01; US: $p$'s < .001), and significantly greater fear of lizards than turtles (HK: $p < .001$; US: $p = .001$).

For children's fear of animals (Table 11 and Fig 3), we found a significant main effect of animal, $F(3, 177) = 25.63$, $p < .001$, $η² = .30$, with children reporting more fear of snakes and spiders than lizards and turtles (all $p$'s < .001, S22 Table in S1 File). We also found a significant site by animal interaction, $F(3, 177) = 34.67$, $p < .001$, $η² = .37$ (S22 Table in S1 File). For between-site comparisons for each animal, US children were more fearful than HK children of snakes ($p < .001$)

**Table 9. Descriptives (Fear Beliefs Data).**

| Speaker | Animal | Overall Sample | | HK | | US | |
|---|---|---|---|---|---|---|---|
| | | Mean | SD | Mean | SD | Mean | SD |
| Children | Snakes | 3.26 | .84 | 2.82 | .41 | 3.71 | .93 |
| | Spiders | 3.22 | .88 | 2.66 | .33 | 3.76 | .90 |
| | Lizards | 2.50 | .94 | 2.71 | .51 | 2.29 | 1.21 |
| | Turtles | 2.48 | .86 | 3.00 | .57 | 1.96 | .78 |
| Parents | Snakes | 4.00 | .83 | 4.29 | .65 | 3.72 | .90 |
| | Spiders | 3.40 | .91 | 3.55 | .98 | 3.26 | .84 |
| | Lizards | 2.55 | .90 | 2.88 | .90 | 2.22 | .77 |
| | Turtles | 1.64 | .48 | 1.60 | .54 | 1.68 | .41 |

*Note.* Children's and parents' fear beliefs scores were measured using different measures but the scoring scale and interpretation is the same (higher average scores indicate greater fear beliefs).

**Table 10. Results for 2(site: US, HK) x 4(animal: snakes, spiders, lizards, turtles) ANOVA on Parents' Fear of Animals.**

| Predictor | df | F | P | η² |
|---|---|---|---|---|
| Between-Subjects | | | | |
| Site | 1 | 7.76 | .01* | .12 |
| Error | 58 | | | |
| Within-Subjects | | | | |
| Animal | 3 | 164.41 | <.001** | .74 |
| Site x Animal | 3 | 3.89 | .01* | .06 |
| Error | 174 | | | |

*p < .05 \* p < .001 \*\*.*

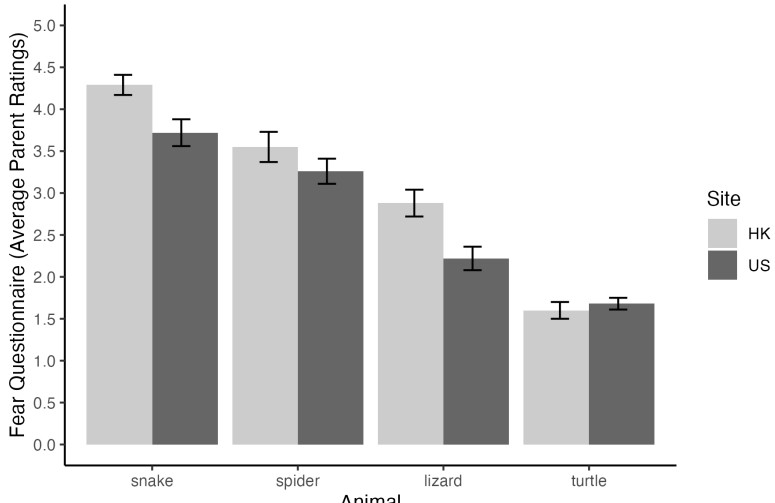

**Fig 2. Results for 2(site: US, HK) x 4(animal: snakes, spiders, lizards, turtles) ANOVA on Parents' Fear of Animals.** Fig 2. represents parents' fear beliefs toward each animal included in the study. The Y axis represents scores on parents' measure of fear and the X axis represents each animal category. The bars represent the average score by site (HK or US) with standard error bars.

and spiders ($p < .001$), and HK children were more fearful of turtles than US children ($p < .001$). For within-site comparisons for animals, US children were significantly more fearful of snakes and spiders than lizards and turtles ($p$'s $< .001$). There were no other significant comparisons for the US sample and no significant comparisons for the HK sample at $p < .05$ (S22 Table in S1 File).

### Language and fear

As additional exploratory analyses, we next explored whether parents' fear was correlated with children's fear of each animal category (S23 Table in S1 File). Across the overall sample, only parents' fear of lizards was significantly and positively correlated with children's fear of lizards ($r = .30$, $p = .01$). Parents' fear of snakes, spiders, and turtles were not significantly correlated with children's fear of snakes, spiders, and turtles, respectively, at $p < .05$. We also explored these relations by site, and nothing was significant at $p < .05$ (S23 Table in S1 File).

We next explored whether parents' use of negative language was correlated with children's use of negative language about each animal (S24 Table in S1 File). Across the overall sample, parents' use of negative language was positively

**Table 11. Results for 2(site: US, HK) x 4(animal: snakes, spiders, lizards, turtles) ANOVA on Children's Fear of Animals.**

| Predictor | df | F | p | η² |
|---|---|---|---|---|
| Between-Subjects | | | | |
| Site | 1 | 1.38 | .24 | .02 |
| Error | 59 | | | |
| Within-Subjects | | | | |
| Animal | 3 | 25.63 | <.001** | .30 |
| Site x Animal | 3 | 34.67 | <.001** | .37 |
| Error | 177 | | | |

*p*<.05 * *p*<.001 **.

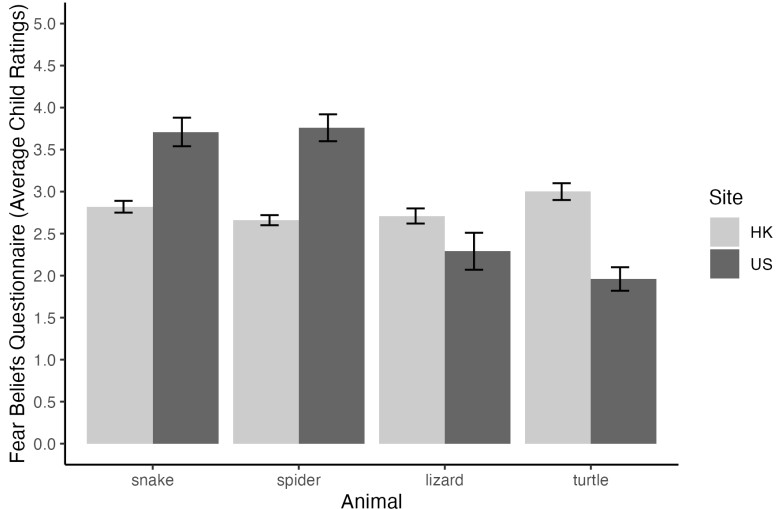

**Fig 3. Results for 2(site: US, HK) x 4(animal: snakes, spiders, lizards, turtles) ANOVA on Children's Fear of Animals.** Fig 3. represents children's fear beliefs toward each animal included in the study. The Y axis represents scores on children's Fear Beliefs Questionnaire and the X axis represents each animal category. The bars represent the average score by site (HK or US) with standard error bars.

correlated with children's negative language for each animal (p's<.001). We also explored these relations by site, and the results were similar with the exception that use of negative language about lizards between parents and children from the HK sample was no longer significant at *p*<.05 (S24 Table in S1 File).

Next, given that parents and children used more negative language during conversations about snakes and spiders and were also more fearful of these animals, we explored whether parents' use of negative language about snakes and spiders predicted children's fear of snakes and spiders. Parents' use of negative language about snakes did not predict children's fear of snakes, $F(1, 59) = .02$, $p = .88$ (S25 Table in S1 File), and parents' use of negative language about spiders did not predict children's fear of spiders, $F(1, 58) = 2.10$, $p = .15$ (S26 Table in S1 File). We further explored site differences by adding site to each model. For snakes, the overall model was significant, $F(3, 57) = 7.81$, $p<.001$ (S27 Table in S1 File). Site was a significant predictor, in which children in the US reported higher levels of fear than HK children regardless of parents' negative language about snakes ($b = 0.98$, $SE = 0.23$, $t = 4.17$, $p<.001$). Neither parents' negative language about snakes nor the interaction between negative language and site were significant at $p<.05$. For spiders, the overall model was significant, $F(3, 56) = 14.01$, $p<.001$ (S28 Table in S1 File). Site was a significant predictor, in which children

in the US reported higher levels of fear than HK children regardless of parents' negative language about spiders (b = 0.93, SE = 0.23, t = 4.03, $p < .001$). Neither parents' negative language about spiders nor the interaction between negative language and site were significant at $p < .05$.

Finally, we explored whether parent report of children's frequency with nature visits was correlated with children's fear beliefs about each animal (S29 Table in S1 File) and their total and proportion (S30 Table in S1 File) use of negative language about each animal. For children's frequency with nature visits and fear beliefs (S29 Table in S1 File), there was only a significant correlation for turtles ($p = .04$) and lizards ($p = .01$), indicating that having more frequent experiences with nature was positively associated with greater fear of turtles and lizards. For children's frequency with nature visits and use of negative language about each animal, nothing was significant when we looked at total or proportion scores (all p's > .14, S30 Table in S1 File).

## General discussion

The current study examined whether parents and children in HK and the US use more negative language during conversations about snakes and spiders than less commonly feared animals like lizards and turtles. We found that parents in both sites used more negative language about snakes and spiders than lizards and turtles, consistent with findings from previous literature [15,16]. We also found that parents and children from both sites expressed more fear of snakes and spiders than lizards and turtles. However, the most interesting finding here is that while HK parents used significantly more negative language about snakes and reported more fear of snakes than US parents, US children reported being more fearful of snakes than the HK children.

What could explain these seemingly contradictory findings? At least for between-site differences among parents, one factor might be US parents' greater prior knowledge of the animals shown in our study compared to HK parents, with the animals being correctly identified by US parents more often overall. In fact, while the US parents could accurately identify some of the snakes and spiders (2 snakes, 3 spiders, 3 lizards, and 3 turtles identified by ≥ 1 parent), the HK parents were nearly at floor (1 snake, 1 spider, and 2 turtles identified by ≥ 1 parent). One explanation for this may be the selection of species included in the study. While animal species were selected from across the globe, a greater number of species included in the study were native to North America and may therefore have been more familiar to parents in the US than HK parents. However, most families in both samples reported living in suburban or urban areas, where direct or everyday experiences with any of these animals may still be rare. Another possible explanation may be how participants were recruited. Participants from the US sample were recruited primarily through online platforms for child research studies, while the HK participants were primarily recruited through a family-targeted eco-tourism company, which may reflect differences in families' prior experiences. Because of these factors, knowledge and attitudes about these animals are likely formed through indirect experiences such as through books, television, or other media sources, which may make opportunities to learn about animals outside of one's environment more common.

Additionally, US parents reported that their children experienced nature—both through nature visits and visits to the zoo or aquarium—more often than HK parents. Together this suggests that the US parents might have more experiences with or are more knowledgeable about these animals in general. Having less experience or connectedness with nature is known to amplify negative attitudes towards nature and wildlife [37,38], and fears towards snakes and spiders in particular [39]. In the case of snakes, more informed knowledge about their biology or life history can lead to more positive attitudes [40–43]. Similarly, people with past, curated interactions with snakes tend to display fewer negative attitudes and less fear towards them [44,45]. Thus, the US parents seemingly had more experience with snakes and with nature in general, possibly leading to less fear. Future studies should better control for or directly explore the impact of ecological relevance of the animals as well as prior learning experiences as they relate to knowledge and attitudes of animals across (and even within) cultures.

Moreover, the felt exposure to snake bite risks by region could also be driving the higher fear towards and incidence of negative language about snakes in HK parents. Interestingly, the only animal correctly identified by more than half of the

HK parents in our study was the highly venomous king cobra, which also happens to be the only animal presented that is native to Hong Kong. In fact, Hong Kong and the surrounding area hosts 47 different snake species, 14 of which are venomous and 8 of which can deliver a fatal bite [46]. As a highly urbanized society interspersed with natural landscapes such as forests and mountains, snakes often inhabit areas adjacent to densely populated areas, resulting in potential conflicts with people [47]. Although again, fatal envenomations are rare, snake bite incidents are more common, and continue to be a medical concern in the region [48–51]. While snake species diversity (e.g., [52]) and snake bite incidents [53,54] are actually higher in some parts of the US, this is mainly limited to southern states. The concern for snake bites could therefore be more prominent in Hong Kong overall given its higher population density, and relatively larger proportion of people living in close proximity to snakes [47].

Overall, while these possibilities support the trends we have observed in parents, our results suggest the opposite in their children. Interestingly, despite the higher incidences of negative language about snakes with HK parents relative to those in the US, we found that US children in our study were more fearful of snakes than HK children. This result contradicts our expectations based on our findings in Hong Kong about higher negative language use in parents and less experience with nature and snakes in children, as well as past work showing adults in Hong Kong to be more fearful of fear-relevant animals than adults in the US [18]. One potential explanation for this finding could be the result of cultural differences in emotional expression. Indeed, previous work has demonstrated that negative emotions are less likely to be expressed in children with Chinese than with American cultural orientations [55,56]. One study found that Hong Kong Chinese middle school children rely less on talking with someone as a regulation strategy for sadness, anger, or fear than their European American counterparts [57]. There is even evidence that this begins in infancy; another study reported that 11-month-old Chinese infants are less facially expressive than European American infants in response to fear-inducing stimuli [58]. Overall, through parent-child conversations, European American parents tend to place a stronger emphasis on the internal, through coaching and teaching their children to express, discuss, and regulate their emotions. Chinese parents, on the other hand, focus less on internally addressing emotions in conversations with their children, and more on external behavior and conduct for bringing an awareness to others' emotions and maintaining group harmony [55,59]. Future work should therefore look to explore alternative methodologies for assessing snake fears in children that better account for cross-cultural differences in how negative emotions are expressed.

Additionally, negative language provided to children was not related to their fear of snakes or spiders. One potential reason for this lack of association might be due to children's prior (both direct and indirect) experiences with these animals. Previous research has documented that preschool age children already know negative information about snakes and spiders [16]. While we did not assess children's prior knowledge of snakes and spiders, in our study, 74% of children had seen a snake and 85% of children had seen a spider prior to participation in the study. This suggests that children have already had some experiences with snakes and spiders prior to this study, and it is possible that such interactions, in conjunction with the conversations surrounding them, may have already shaped children's attitudes of snakes and spiders. Future studies can more thoroughly explore parents' and children's prior knowledge and experiences of snakes and spiders, and how emotional input can alter children's already formed attitudes of snakes and spiders. However, a recent study of children from the US has demonstrated that hearing information about snakes in a neutral or anthropomorphic way in a single intervention can reduce children's fear of snakes, and that anthropomorphizing how children hear about snakes may also improve their attitudes and willingness to help snakes [17]. This opens the door for further exploration of the impact of brief interventions on children's attitudes toward commonly feared animals across the globe.

Thus, one of the limitations for this current study is that we assessed snake fears in children through verbal communication, which could have hindered emotional expression in HK children. We therefore recommend future cross-cultural research assessing animal fears in children to consider non-verbal approaches. In further expanding this research cross-culturally, future studies should also select a balanced set of animal stimuli representative of all geographic regions sampled. In this study, 2 snakes could be found in North America, 1 in Asia, 1 in Africa, and overall, 5 of the 16 animals

could be found in North America, while 2 were from Asia. While most parents in our study were able to identify the king cobra, accuracy for the other animals was especially low for the Hong Kong sample. The recognition of snakes in particular may vary based on local significance in the context of venomous species and snakebite risk [60,61]. Including animals from different continents around the world assesses the recognition of animals commonly featured in media but should be complemented with more locally relevant species to even out rates of parents' knowledge of the animals presented.

To our knowledge, this study is the first to compare parent-child conversations about snakes (and spiders, lizards, and turtles) between cultures, and the potentially resulting effects on snake fears. Our results suggest that negative language use about snakes and fear towards snakes can be communicated differently from parent to child across cultural contexts—at least as far as the outward expression of fear is concerned. These findings highlight the influence of how parents and educators speak to children about commonly feared animals, and how resulting attitudes towards these animals by children may manifest differently according to place, geography and culture. Conservation outreach and educational efforts addressing children's attitudes towards wildlife, and their fears towards snakes in particular, should consider the social and natural environments in which they are raised for a culturally-nuanced understanding of the younger generation's relationships with the natural world.

## Supporting information

**S1 File. This document provides all supplemental tables and figured referenced in the manuscript.**
(DOCX)

## Acknowledgments

We would like to thank all parents and children who participated and a special thank you to Little Woods Nature Education Limited for their help in the recruitment of participants.

## Author contributions

**Conceptualization:** Reider Lori B., Landry Yuan Félix, LoBue Vanessa.

**Data curation:** Reider Lori B., Landry Yuan Félix, Leung Even Y.M..

**Formal analysis:** Reider Lori B..

**Funding acquisition:** LoBue Vanessa.

**Investigation:** Reider Lori B., Landry Yuan Félix, Leung Even Y.M., Yeung Karen K.L., Leung Samantha Hing Lam, Hai Catherine Wai Ching, Lei Janet Hiu Ching.

**Methodology:** Reider Lori B., LoBue Vanessa.

**Project administration:** Reider Lori B., Landry Yuan Félix.

**Supervision:** LoBue Vanessa.

**Visualization:** Reider Lori B..

**Writing – original draft:** Reider Lori B., Landry Yuan Félix, LoBue Vanessa.

**Writing – review & editing:** Reider Lori B., Landry Yuan Félix, Leung Even Y.M., LoBue Vanessa.

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
