## [Decision Letter · Decision Letter 0]

22 Jan 2026

Dear Dr. Reider,

Thank you for submitting your manuscript to PLOS ONE. After careful consideration, we feel that it has merit but does not fully meet PLOS ONE’s publication criteria as it currently stands. Therefore, we invite you to submit a revised version of the manuscript that addresses the points raised during the review process.

We look forward to receiving your revised manuscript.

Kind regards,

Renato Filogonio

Academic Editor

PLOS One

Journal Requirements:

“The current research was funded by a grant by the James S. McDonnell Foundation to PI LoBue.”

Additional Editor Comments:

I agree with the reviewer's assessment that the table and figure captions need to be more informative (see below). Additionally, I think that the 'y' axis from the figures is a little misleading, so perhaps discerning that those are 'scores' would be more informative. Finally, I think that it would be beneficial if the authors explained the difference between 'fear levels' and the 'willingness to approach the animal', as described in the Material and Methods (line 250). As it is, those measurements seem redundant (that is, the more a person fears an animal, the less willing that person is to approach that animal).

Reviewers' comments:

Reviewer's Responses to Questions

**Comments to the Author**

1. Is the manuscript technically sound, and do the data support the conclusions?

Reviewer #1: Yes

Reviewer #2: Partly

2. Has the statistical analysis been performed appropriately and rigorously?

Reviewer #1: Yes

Reviewer #2: Yes

3. Have the authors made all data underlying the findings in their manuscript fully available?

Reviewer #1: Yes

Reviewer #2: No

4. Is the manuscript presented in an intelligible fashion and written in standard English?

Reviewer #1: Yes

Reviewer #2: Yes

Reviewer #1: The present manuscript examines how parent-child conversations about different animals might shape children’s fear beliefs in the United States and Hong Kong. The study recruited 62 dyads with a 4- to 6-year-old child across the two countries. They found that parents and children had more negative conversations about spiders and snakes than turtles and lizards, suggesting that these conversations might predict fear beliefs (as those were also the animals people feared the most). There was no concurrent relation between fear beliefs and conversation affect. The study was pre-registered, although an entire location was excluded from data collection due to logistical considerations. The study is, for the most part, methodologically sound, and the data are interpreted accurately. Below, I include some comments I hope will strengthen the manuscript.

1. One aspect that the manuscript could pay more attention to is the familiarity of the animal species. The manuscript details how US parents recognized more animals than Hong Kong parents, but does not connect this to the habitat of those animals. For example, garter and bull snakes, brown recluse spiders, Gila monsters, and alligator snapping turtles are found in North America, potentially making them more familiar to participants from the United States. Jumping spiders and skinks are found worldwide and thus potentially common for all participants. But only the king cobra was native to Hong Kong. This familiarity could influence the results and the conversations that parents and children had. For example, the greater familiarity could explain the difference between the US and Hong Kong in fear beliefs.

2. The authors discuss in their introduction how exposure to venomous snakes is not even throughout the world. One point that could be highlighted more is that even within a country, there is unevenness in terms of who is at greater risk of coming into contact with these animals. This is of particular importance because the sample characteristics suggest that the families that participated in this study are likely at a low risk, as they are mostly in urban areas. Additionally, recruiting participants in Hong Kong through an eco-tourism company might have also influenced the results.

3. I also wonder if children’s experiences with nature are related to their fear beliefs or their negative talk.

4. I think the manuscript could provide a greater rationale for why the 4- to 6-year-old age range was selected for this study. Why are these ages interesting for the development of fear beliefs or to examine conversations?

5. I also believe the authors can discuss more about how, in their sample, negative talk was not related to children’s fear beliefs. Given that it was argued that these conversations are one mechanism for the development of these fears, how does this lack of an association change our understanding (or not) of how these fears develop? Are these fear beliefs malleable enough that we should expect changes after one book reading session?

Reviewer #2: The topic of the study is very interesting, and research in this area is always recommended. However, I believe the manuscript could be improved in terms of fluency and theoretical grounding, in order to make it suitable for publication in this journal.

Below are some suggestions.

1 - Introduction: Overall, I suggest a revision of the introduction. The information presented is interesting, but it could be written more concisely. Some sentences that link to subsequent statements could be shortened rather than maintaining multiple long, similar sentences. Additionally, I recommend a clearer separation between the topics of spiders and snakes. At times, the information on spiders and snakes is so intermixed that it seems mistaken.

2- Line 50: Please provide the source for this information and include appropriate references. This information may also vary across cultural contexts (e.g., Eastern and Western cultures).

3- Line 64 - reference 8: Is an article focused on snakes the most appropriate reference for a global discussion of venomous spiders?

4- Line 64: It lacks a better connection between sentences.

5- Line 73-75: However, the geography of each region should also be considered, as rural areas exhibit higher encounter rates than urban areas. This may serve as a counterargument to using the mean as a representative value

6- Line 91-92: These data are interesting, as these other animals are also not grouped into the cute or charismatic fauna.

7- Line 132-136: It is worth providing a more focused explanation of why these two countries were selected.

8- Line 153-155: Please re-order. These should be final sentences of the section.

9- Line 158: Collected how?

10- Line 164: If it’s just one, you can specify what it is

11- Line 212-215: It would be helpful to include a link or a screenshot of the page for better understanding

12- Table 1: The table caption is ambiguous, as it is unclear whether ‘threat relevance’ refers to conservation status or to the potential threat posed to humans.

The caption should be self-explanatory

13- Table 4: The caption could be improved by including additional information, such as the total number of participants in each case.

14- Line 436: Caption can be improved

15- Line 447: Same comment

16- Tables and figures in general: I recommend revision and improvement of all the captions. Subtitles must be informative on their own

17- Line 574-577: In my opinion, many parts of the Results section could be rewritten in a more fluid way. For example, in this case, the comparison could be presented using a linear order of fear, rather than repeating each item one by one.

18- Line 638-643: It is possible that I am missing a distinction here, but this passage appears repetitive, as similar comparisons are presented across consecutive sentences.

19- Line 646-660: Regarding the conclusion of this paragraph, I have a few concerns. First, the greater access to and knowledge of animals reported for U.S. parents may also reflect cultural and/or socioeconomic differences between sites, which are not explicitly discussed. Second, such cultural factors could influence attitudes toward reptiles for both parents and children. Finally, it is possible that I am misunderstanding the argument, but the last sentence may be somewhat confusing: while it suggests that greater experience and knowledge among U.S. parents is associated with lower fear, the comparison with children could give the opposite impression. To avoid this potential confusion, this might be a good place to discuss the transmission of knowledge, attitudes, and fear from parents to children, and how these relationships may shape children’s responses.

.

Reviewer #1: **Yes:**David MenendezDavid MenendezDavid MenendezDavid Menendez

Reviewer #2: No

---

## [Author Response · Author response to Decision Letter 1]

9 Mar 2026

Response to Reviewers

Thank you for the thoughtful feedback and the opportunity to revise and resubmit our manuscript.

General Comments:

All PLOS ONE’s style requirements should now be met.

“The current research was funded by a grant by the James S. McDonnell Foundation to PI LoBue.”

We have now included this information in our cover letter.

All data are now available through Databrary (https://databrary.org/), a restricted access web-based data library specialized for storing and sharing research data. The link to the specific study volume is: https://databrary.org/volume/1407 . Data may also be accessed through Open Science Framework using the following link: https://osf.io/2rp4j/overview?view_only=ecab827fdd2f4fb8a676dd58ff260daa .

We have now added captions and/or updated titles of tables and figures to improve clarity.

We were not asked to cite specific publications by the reviewers or editor.

Additional Editor Comments:

I agree with the reviewer's assessment that the table and figure captions need to be more informative (see below). Additionally, I think that the 'y' axis from the figures is a little misleading, so perhaps discerning that those are 'scores' would be more informative. Finally, I think that it would be beneficial if the authors explained the difference between 'fear levels' and the 'willingness to approach the animal', as described in the Material and Methods (line 250). As it is, those measurements seem redundant (that is, the more a person fears an animal, the less willing that person is to approach that animal).

Thank you for this feedback. First, we have updated the table and figure captions throughout the manuscript so they are more informative to the reader. Second, we have updated the y axis for Figure 2 (Fear Questionnaire (Average Parent Ratings)) and Figure 3 (Fear Beliefs Questionnaire (Average Child Ratings)) to be more informative. Further, we have clarified that the three items asked of parents collectively measure parents’ fear and 2 of the 3 items were sampled from the child questionnaire (and as the editor notes, both reporting high fear and low willingness to approach would collectively indicate higher levels of fear).

Reviewer #1:

The present manuscript examines how parent-child conversations about different animals might shape children’s fear beliefs in the United States and Hong Kong. The study recruited 62 dyads with a 4- to 6-year-old child across the two countries. They found that parents and children had more negative conversations about spiders and snakes than turtles and lizards, suggesting that these conversations might predict fear beliefs (as those were also the animals people feared the most). There was no concurrent relation between fear beliefs and conversation affect. The study was pre-registered, although an entire location was excluded from data collection due to logistical considerations. The study is, for the most part, methodologically sound, and the data are interpreted accurately. Below, I include some comments I hope will strengthen the manuscript.

1. One aspect that the manuscript could pay more attention to is the familiarity of the animal species. The manuscript details how US parents recognized more animals than Hong Kong parents, but does not connect this to the habitat of those animals. For example, garter and bull snakes, brown recluse spiders, Gila monsters, and alligator snapping turtles are found in North America, potentially making them more familiar to participants from the United States. Jumping spiders and skinks are found worldwide and thus potentially common for all participants. But only the king cobra was native to Hong Kong. This familiarity could influence the results and the conversations that parents and children had. For example, the greater familiarity could explain the difference between the US and Hong Kong in fear beliefs.

Thank you for this thoughtful comment. We agree that knowledge of and attitudes toward the different animals may be influenced by the ecological relevance of the animal to where one lives. We have now included this as a limitation of the study in the discussion section in lines 630 – 653.

2. The authors discuss in their introduction how exposure to venomous snakes is not even throughout the world. One point that could be highlighted more is that even within a country, there is unevenness in terms of who is at greater risk of coming into contact with these animals. This is of particular importance because the sample characteristics suggest that the families that participated in this study are likely at a low risk, as they are mostly in urban areas. Additionally, recruiting participants in Hong Kong through an eco-tourism company might have also influenced the results.

We agree with this point. To address it, we have added the following sentence to the introduction in lines 69-70: “Even within these broad regions of the world, the prevalence of encounters with snakes varies based on ecological relevance and livelihood. We have also addressed this in the General Discussion (see response to reviewer 1, comment 1).

3. I also wonder if children’s experiences with nature are related to their fear beliefs or their negative talk.

Thank you for this interesting question. To explore these ideas, we ran Spearman’s rank correlations to explore whether parent report of children’s frequency with nature visits was correlated with children’s fear beliefs about each animal and their use of negative language about each animal. For children’s fear, there was no significant link for snakes (p = .65) or spiders (p = .28). There was a significant correlation for turtles (p = .04) and lizards (p = .01), indicating that having more frequent experiences with nature was positively associated with greater fear of turtles and lizards. For children’s frequency with nature visits and use of negative language about each animal, nothing was significant when we looked at raw or proportion scores (all p’s > .14). We have now added this to the manuscript in the analysis section, lines 608 - 615. Further, we now also acknowledge this as an area for future research on page 31.

4. I think the manuscript could provide a greater rationale for why the 4- to 6-year-old age range was selected for this study. Why are these ages interesting for the development of fear beliefs or to examine conversations?

We selected this age range as this is when animals fears and phobias begin to emerge as well as the developmental skills required to complete the tasks in the study. We have now clarified this in the participants section: This age range was selected as this is around the time when animal fears and phobias first emerge in childhood [31].

5. I also believe the authors can discuss more about how, in their sample, negative talk was not related to children’s fear beliefs. Given that it was argued that these conversations are one mechanism for the development of these fears, how does this lack of an association change our understanding (or not) of how these fears develop? Are these fear beliefs malleable enough that we should expect changes after one book reading session?

We have now added a paragraph to the General Discussion to unpack this lack of association in our findings in lines 689 – 705, Additionally, negative language provided to children was not related to their fear of snakes or spiders. One potential reason for this lack of association might be due to children’s prior (both direct and indirect) experiences with these animals. Previous research has documented that preschool age children already know negative information about snakes and spiders [16]. While we did not assess children’s prior knowledge of snakes and spiders, in our study, 74% of children had seen a snake and 85% of children had seen a spider prior to participation in the study. This suggests that children have already had some experiences with snakes and spiders prior to this study, and it is possible that such interactions, in conjunction with the conversations surrounding them, may have already shaped children’s attitudes of snakes and spiders. Future studies can more thoroughly explore parents’ and children’s prior knowledge and experiences of snakes and spiders, and how emotional input can alter children’s already formed attitudes of snakes and spiders. However, a recent study of children from the US has demonstrated that hearing information about snakes in a neutral or anthropomorphic way in a single intervention can reduce children’s fear of snakes, and that anthropomorphizing how children hear about snakes may also improve their attitudes and willingness to help snakes [17]. This opens the door for further exploration of the impact of brief interventions on children’s attitudes toward commonly feared animals across the globe.

Reviewer #2:

The topic of the study is very interesting, and research in this area is always recommended. However, I believe the manuscript could be improved in terms of fluency and theoretical grounding, in order to make it suitable for publication in this journal. Below are some suggestions.

1 - Introduction: Overall, I suggest a revision of the introduction. The information presented is interesting, but it could be written more concisely. Some sentences that link to subsequent statements could be shortened rather than maintaining multiple long, similar sentences. Additionally, I recommend a clearer separation between the topics of spiders and snakes. At times, the information on spiders and snakes is so intermixed that it seems mistaken.

Thank you for this suggestion. We have edited the introduction significantly to make it more concise, and eliminated spiders from the argument completely, since the study centers around snakes. We hope this improves the readability of the manuscript.

2- Line 50: Please provide the source for this information and include appropriate references. This information may also vary across cultural contexts (e.g., Eastern and Western cultures).

We have rephrased the opening lines and included appropriate references.

3- Line 64 - reference 8: Is an article focused on snakes the most appropriate reference for a global discussion of venomous spiders?

Thank you for catching this. We have now focused the introduction on snakes, given their cultural significance for our sample.

4- Line 64: It lacks a better connection between sentences.

We have now reworded the sentence to read: While snake envenomation is more common globally, fatal encounters are rare, with an estimated 1.2 to 5.5 million people envenomed by snakes and about 125,000 (2% - 10%) deaths annually [10].

5- Line 73-75: However, the geography of each region should also be considered, as rural areas exhibit higher encounter rates than urban areas. This may serve as a counterargument to using the mean as a representative value

We agree that geographical location (and thus the likelihood of direct experiences with snakes) is important for developing knowledge and attitudes about snakes, but also acknowledge that fear of snakes is quite common across the globe. We have reworded this sentence to read: “Thus, while some snake species do constitute a threat to modern-day humans in many parts of the world, snake fears remain widespread across the globe regardless of experiential or ecological relevance to humans.” We also revisit this in the discussion (see response to Reviewer 1, comment 1) as a limitation of our study and important area for future research.

6- Line 91-92: These data are interesting, as these other animals are also not grouped into the cute or charismatic fauna.

We agree, and there are several papers using animals like frogs, turtles, and lizards as comparison to stimuli of snakes and spiders among children’s and adults’ fear and rapid detection of these animals, given some of the similar physical features of these animals as well as their commonality together such as in the Zoo, in picture books, etc.

7- Line 132-136: It is worth providing a more focused explanation of why these two countries were selected.

We now include an explicit sentence in the Introduction about why these two regions were chosen, and we provide substantial information about how cultural beliefs about snakes differs between the regions: “We chose these two regions because they reflect different worldviews about, and may have potentially different experiences with snakes, providing a meaningful basis to examine how cultural contexts may influence parent-child conversations about snakes, and animals more broadly.

8- Line 153-155: Please re-order. These should be final sentences of the section.

We have moved the first two sentences to the end of the paragraph.

9- Line 158: Collected how?

We have clarified this sentence to include how data were collected: All data were collected during an online Zoom call using Qualtrics during the COVID-19 pandemic, from February 9 through May 26, 2022.

10- Line 164: If it’s just one, you can specify what it is

In our survey, there was the option to write in their relationship if “Other” was selected, and the one caregiver who selected “Other” omitted a text response so we cannot specify the specific relationship to the child using the survey data.

11- Line 212-215: It would be helpful to include a link or a screenshot of the page for better understanding

We are unable to publish a sample of the actual book pages due to copyright restrictions with the images used in the study. However, we have added S1 Fig to include a sample layout of the picture book page for reference.

12- Table 1: The table caption is ambiguous, as it is unclear whether ‘threat relevance’ refers to conservation status or to the potential threat posed to humans.

The caption should be self-explanatory

We have updated the title of Table 1 to read: Table 1. List of animal species included i

---

## [Decision Letter · Decision Letter 1]

7 Apr 2026

From Words to Worries: A Cross-Cultural Comparison of Parent-Child Conversations About Snakes in Early Childhood

PONE-D-25-55348R1

Dear Dr. Reider,

We’re pleased to inform you that your manuscript has been judged scientifically suitable for publication and will be formally accepted for publication once it meets all outstanding technical requirements.

Kind regards,

Renato Filogonio

Academic Editor

PLOS One

Additional Editor Comments (optional):

Reviewers' comments:

Reviewer's Responses to Questions

**Comments to the Author**

Reviewer #1: All comments have been addressed

2. Is the manuscript technically sound, and do the data support the conclusions?

Reviewer #1: Yes

3. Has the statistical analysis been performed appropriately and rigorously?

Reviewer #1: Yes

4. Have the authors made all data underlying the findings in their manuscript fully available?

Reviewer #1: Yes

5. Is the manuscript presented in an intelligible fashion and written in standard English?

Reviewer #1: Yes

Reviewer #1: The authors have addressed all of my comments from my previous review, and I do not have any further comments.

.

Reviewer #1: **Yes:**David MenendezDavid MenendezDavid MenendezDavid Menendez

---

## [Editor Report · Acceptance letter]

PONE-D-25-55348R1

PLOS One

Dear Dr. B.,

I'm pleased to inform you that your manuscript has been deemed suitable for publication in PLOS One. Congratulations! Your manuscript is now being handed over to our production team.

Kind regards,

on behalf of

Dr. Renato Filogonio

Academic Editor

PLOS One